# Exploring the Relationship between Anemia and Postpartum Depression: Evidence from Malawi

**DOI:** 10.3390/ijerph20043178

**Published:** 2023-02-11

**Authors:** Zijing Cheng, Mahesh Karra, Muqi Guo, Vikram Patel, David Canning

**Affiliations:** 1Department of Global Health and Population, Harvard T.H. Chan School of Public Health, Boston, MA 02115, USA; 2Department of Public Health Sciences, University of Rochester Medical Center, Rochester, NY 14642, USA; 3Frederick S. Pardee School of Global Studies, Boston University, Boston, MA 02215, USA; 4Department of Global Health and Social Medicine, Harvard Medical School, Boston, MA 02115, USA

**Keywords:** postpartum depression, Anemia, Malawi, PHQ-9, major depressive disorder

## Abstract

Purpose: Study findings suggest association between anemia and postpartum depression, but available evidence is scant and inconsistent. We investigate whether anemia is related to postpartum depression among women who have recently given birth in Malawi, where anemia prevalence is high. Methods: We use cross-sectional data from 829 women who were 18–36 years old, married, lived in Lilongwe, Malawi, and gave birth between August 2017 and February 2019. The primary outcome is postpartum depression in the year after birth, defined by the Patient Health Questionnaire-9 (PHQ-9). Anemia status was assessed using hemoglobin levels that were measured at the time of the interview. Multivariate logistic regression analyses were used to investigate the relationship between postpartum depression and anemia status. Results: Our analysis sample consists of 565 women who completed the PHQ-9, tested for anemia, and had no missing values for covariates. Of these women, 37.5% had anemia (hemoglobin levels ≤ 110 g/L), and 2.7% were classified as showing symptoms of a major depressive disorder (MDD). After adjusting for potential confounders, anemia was significantly associated with increased risk of MDD (OR: 3.48, 95% CI: 1.15–10.57, *p*-value: 0.03). No significant associations were found between other covariates and postpartum depression. Conclusions: Our findings suggest a potential association between anemia and postpartum depression among women in Malawi. Policies that aim to improve nutrition and health outcomes for pregnant and postpartum women could generate a “double benefit” by both preventing anemia and reducing the risk of postpartum depression.

## 1. Introduction

Postpartum depression is one of the most common mental disorders in the postnatal period, affecting an estimated 10–15% of mothers worldwide; in Malawi, our study setting, 11% of new mothers are estimated to be affected by postpartum depression [1,2]. According to the World Health Organization, major depressive disorder (MDD) is identified as one of the major causes to global burden of diseases, with a disability adjusted life year (DALY) weighting of 0.66, meaning that a year of life with depression has the same value of 0.34 years of life in full health [3].

Anemia during pregnancy is a serious public health issue in low- and middle-income countries. In Malawi, an estimated 41.8 percent of pregnant women in 2016 were classified to be anemic (having a hemoglobin level less than 110 g per liter) [3]. Fortunately, anemia has a relatively low global burden of disease, with disability weights of 0.004, 0.052, and 0.149 for mild, moderate, and severe anemia, respectively, and can be treated with a modified diet, particularly through increased iron supplementation [3].

Among symptoms that are often observed in postpartum depressed women, anemia is an important concern because both conditions share common features, ranging from tiredness and dizziness to irritability [4]. Anemia is much more common in low-income settings, a factor which may explain why low-income countries have higher rates of post-partum depression than high income countries, with particularly high rates in Africa [5,6]. Several studies have previously examined the potential associations between anemia and postpartum depression and have documented mixed findings. A recent meta-analysis found that anemia during and after pregnancy significantly increased the risk of postpartum depression [7]. Moreover, in studies from Japan, Iran, and Saudi Arabia, anemia was shown to be associated with postpartum depression [8,9,10]. In contrast, null or inverse associations were reported in studies conducted in China and Sweden [11,12]. Despite of the high postpartum depression rates and high prevalence of anemia in Malawi, empirical evidence in these regards is largely lacking, with only few recent studies exploring potential risk factors of antenatal depression among certain groups and regions in Malawi [13,14].

Motivated by these inconsistent findings and by the existing gaps in the literature, we collected individual-level data among Malawian women and examined the relationship between their anemia status and postpartum depression. Specifically, we hypothesized that anemic mothers would be associated with higher risk of experiencing postpartum depression.

## 2. Materials and Methods

### 2.1. Setting and Participants

In this study, we investigated the relationship between anemia and postpartum depression in a sample of women in urban Malawi. The data in this study were obtained as part of a larger randomized controlled trial that primarily aimed to understand the use of family planning and reproductive health services in Lilongwe, the capital of Malawi [15]. The inclusion criteria were that, at the time of the baseline survey, women (1) were married; (2) were either currently pregnant or had given birth within the prior six months; (3) were between the ages of 18 and 35; and (4) lived in the city of Lilongwe.

As part of the trial, a baseline survey was conducted, with a sample of 2143 pregnant and immediately postpartum women in 2016, and two follow-up surveys were conducted annually in 2017 and 2018. For this study, we included 829 women who were married, lived in the city of Lilongwe, had given birth between August 2017 and February 2019, and had completed the screening questionnaires for postpartum depression as part of the two follow-up surveys that were conducted in 2017 and 2018; data on postpartum depression were not collected as part of the baseline questionnaire.

### 2.2. Assessment of Depressive Symptoms

Postpartum depression generally occurs within four weeks of giving birth and even as late as 7.5 months postpartum [16]. In our study, we defined the postpartum period as the period immediately following the birth of a child up to one year after the birth, by which time fertility typically resumes.

As part of the follow-up surveys, women were screened for postpartum depression using the Patient Health Questionnaire-9 (PHQ-9), one of the most widely used screening tools for postpartum depression [17]. This self-report instrument is derived from the Primary Care Evaluation of Mental Disorder and consists of nine items assessing symptoms experienced in the prior two weeks. The PHQ-9 has been shown to be a reliable and valid instrument in a number of settings, including in Malawi where a Chichewa translated version of the instrument has been used in a number of studies [18].

The original PHQ-9 questionnaire is presented in Appendix A. Its total score is a summation of the nine questions, each of which are scored from 0 to 3 points depending on the frequency of symptoms experienced (“not at all” = 0, “several days” = 1, “more than half the days” = 2, and “nearly every day” = 3). We used a cutoff score of 10 or more on the PHQ-9 to detect moderate to severe postpartum depression. In addition, we applied the “diagnostic algorithm” for depression (the DSM-IV criteria) in which, if a respondent was identified to have MDD, the respondent would rate a score of 2 or 3 on at least one of the first two questions and would rate a 2 or a 3 at least four of the remaining items (with the exception that a score of 1 is sufficient for item 9). This “diagnostic algorithm” has been found to have a higher sensitivity (0.88) and specificity (0.80) for identifying MDD than using a cutoff score of 10 on the PHQ-9 [19].

### 2.3. Hemoglobin Assessment

As part of the survey, each enrolled woman was asked to participate and was administered with a rapid test for hemoglobin via capillary blood collection. The hemoglobin measurement was carried out at the same time as the depression questionnaire. Among all women who were asked, a total of 239 women refused to consent to participate in the test. Hemoglobin levels were measured in gram/liter (g/L) using the HemoCue Hb201 point-of-care hemoglobin analyzer, which provided results within one minute of testing [20]. After adjusting for the elevation of Lilongwe, and following the World Health Organization (WHO) guidelines for assessing anemia status in pregnant women, the presence of anemia was defined as having an adjusted hemoglobin level of less than or equal to 110 g/L [21]. For our analysis, we defined the presence of anemia as a binary exposure.

### 2.4. Other Covariates

To minimize potential confounding variables, we also collected data on variables that have been considered to be associated with postpartum depression. These variables include women’s age (defined as four age groups: 15–19; 20–24; 25–29; and 30–39), women’s educational attainment (no schooling and primary; secondary and higher), employment status (working or not), marital status (currently married or living as married, or not), polygamy (whether the woman’s husband has other wives), whether the woman has given multiple births (a binary variable), and the total number of children alive. At the time of interview women were asked to complete the depression questionnaire if they had given birth in the last year. This means that there is variation in months since birth in our data. However, an analysis of 565 studies of post-partum depression found no significant difference in prevalence rates depending on the period of the study post birth (less than 3 months, 3–6 months, 6–12 months). We found months since birth was not a significant predictor of depression in our regression analysis and our results on the effects of anemia on major depressive disorder are robust to the addition of this variable (see Appendix A).

### 2.5. Statistical Analysis

We present descriptive statistics to summarize the characteristics and key outcomes for our analytic sample of 829 women who completed the PHQ-9 questionnaire. A total of 264 women were excluded due to missing information on covariates: 239 women had missing data on anemia status (e.g., they refused to consent for an anemia test), 19 women had missing data on marriage, and 6 women had missing data on educational attainment. To identify potential selection bias due to missing data, we conducted a chi-square test to compare the characteristics of participants and non-participants; no significant differences were found (see Appendix A). The final analysis was conducted on the analytic sample of 565 women for whom complete outcome and covariate information was available.

In order to compare demographic characteristics and outcomes between women with and without postpartum depression, t-tests were used for continuous variables and chi-square analyses were used for categorical variables. Since the outcome variable (the reporting of postpartum depression based on responses to the PHQ-9 questionnaire) is binary, we conducted multivariate binary logistic regression analyses to investigate the relationship between postpartum depression and anemia status, adjusting for the covariates that we identify above. We report adjusted odds ratios (ORs) and 95% confidence intervals (CIs) to quantify the associations of our anemia exposure and three measures of depression. Categories with fewer than ten observations were excluded from the analysis, and confidence intervals were adjusted for heteroskedasticity using Eicker-Huber-White heteroskedastic-robust standard errors. Finally, a Hosmer-Lemeshow test was conducted to evaluate the model’s goodness-of-fit.

All statistical analyses were conducted using STATA (version 16), and a *p*-value of under 0.05 was considered to be statistically significant.

### 2.6. Ethical Approval

Ethical approval to conduct the main randomized controlled trial in Malawi was received from the Harvard University Institutional Review Board (protocol number IRB16-0421) and the Malawi National Health Sciences Research Committee (protocol number 16/7/1628). An informed consent was obtained from all participants who were recruited into the study.

The trial was registered at the American Economic Association Registry for randomized controlled trials on 7 May 2015 (AEARCTR-0000697) and at the Registry for International Development Impact Evaluations (RIDIE) on 28 May 2015 (RIDIE-STUDY-ID-556784ed86956). The complete protocol for the trial was published elsewhere [15]).

## 3. Results

### 3.1. Demographic Characteristics

The selection of the final sample and analytic subsample with complete data is shown in Figure 1. Table 1 presents descriptive statistics for the sample and analytic subsample. From a total of 867 eligible women who were interviewed in the second and third waves of the study, 829 (95.62%) women completed the PHQ-9 questionnaire. Reasons for non-completion included refusal, not being interviewed in person (women who answered an abbreviated phone survey did not respond to the PHQ-9 questionnaire), and missing and uninterpretable data. Women in the final sample were between 18 and 36 years old, with a mean age of 24.9 years; 448 (54.0%) women had no schooling or only had primary education; 527 (63.6%) women were unemployed; 751 (90.6%) reported being in a monogamous marriage; and 811 (97.8%) had no multiple births (twins, triplets, etc.). On average, women in our sample have 2.3 children born at the time of the interviews.

Among those 829 women who completed the PHQ-9 questionnaire, a total of 565 women (68.2%) had no missing data on either anemia status or on other covariates. Key reasons for missing data on anemia and covariates included refusal to participate in a rapid blood test, failure to follow up following the interview, and missing data from key control variables. The final analytic subsample of 565 women were, on average, 24.8 years old. Of these women, 307 (54.3%) had no schooling or only had primary education; 371 (65.7%) were unemployed; 534 (94.5%) reported being in a monogamous marriage; and 549 (97.2%) had no multiple births (twins, triplets, etc.). On average, women in the analytic subsample have 2.3 children.

### 3.2. Anemia Status

Among the women in our analytic subsample for whom hemoglobin measurements were completed, the adjusted hemoglobin level for pregnant women ranged from 60 g/L to 161 g/L, with a mean and standard deviation of 123.4 g/L ± 14.8 g/L. Of the 565 women in our analytic subsample, 212 women (37.5%) were found to be anemic (had hemoglobin levels of ≤110 g/L). Among tested women, 353 (62.5%) women were found to be non-anemic, 120 women (21.2%) were classified as having mild anemia (a level of hemoglobin concentration between 100 g/L to 109 g/L), 87 women (15.4%) were classified as having moderate anemia (hemoglobin levels between 70 g/L and 99 g/L), and 5 women (0.9%) were classified as having severe anemia (hemoglobin levels below 70 g/L). The distribution of hemoglobin levels from the analytic subsample is shown in Figure 2.

### 3.3. Depressive Symptoms

Table 1 also presents descriptive statistics for the prevalence of depression in our sample. The average PHQ-9 score in the sample was 2.2 and ranged from 0 to 24 for the sample. Among the subsample of 565 women, 52.2% of women scored 0 points (no depression), 28.9% scored 1–4 points (minimal depression), 13.8% scored 5–9 points (mild depression), 3.9% scored 10–14 points (moderate depression), 0.5% scored 15–19 points (moderately severe), and 0.7% scored 20–24 points (severe depression). Using the “diagnostic algorithm” as the threshold, it was found that 2.7% of women in our sample were considered as showing symptoms of major depressive disorder (MDD). This prevalence of major depressive disorder is significantly lower than previous findings from Malawi, which estimated 10.7% adjusted weighted prevalence of major depressive episodes among women in Malawi [13], as well as an estimated average weighted prevalence of 18.3% for postnatal depression in Sub-Saharan Africa [22].

When adjusting the cutoff value for major depressive disorder to 10 points, we estimated an adjusted prevalence of 5.1%, which is larger than the estimated prevalence from the “diagnostic algorithm” but still much lower than estimates from other studies. The mean PHQ-9 score among the 15 participants who were diagnosed with major depressive disorder was 15.3, while the mean PHQ-9 score for women who scored greater than or equal to 10 points was 13.4. Specifically, Figure 3 presents the distribution of PHQ-9 scores for the subsample. When comparing raw PHQ-9 scores among women based on their anemia status, we estimated a mean PHQ-9 score of 2.4 for anemic women, compared to a mean PHQ-9 score of 2.1 among non-anemic women. A chi-square test for trends on anemia as an ordinal variable found no significant differences in mean PHQ-9 scores by women’s anemia status (see Table 2).

### 3.4. Multivariate Analyses

Adjusted associations between women’s anemia status and postpartum depression from the multivariate logistic regression models are shown in Table 3. Model 1 presents estimates from the regression where postpartum depression was classified using the “diagnostic algorithm”. Findings from this model suggest that anemia was significantly associated with an increased risk of postpartum depression (OR 3.48, 95% CI 1.15–10.57, *p*-value 0.03). No significant associations were found between other covariates and postpartum depression in this model. Model 2 presents estimates from the regression where postpartum depression was classified using a cutoff score of 10 or higher on the PHQ-9 questionnaire. Neither anemia nor other covariates were found to be significantly associated with the risk of postpartum depression at the 5% level.

In Appendix A, we re-estimated Models 1 and 2 using anemia as an ordinal variable, and we found similar results to those presented in Table 3, especially for mild anemia. Hence, we recognized the relatively small number of observations for estimating associations between our depression outcomes and moderate and severe anemia, a factor which limits our power to precisely estimate the associations for these two categories. To compare the use of anemia as a binary variable (Model 1) and an ordinal variable (Model 3), we further conducted a likelihood-ratio test between the models and noted that there is not enough evidence to suggest that Model 3 is a better fit than Model 1.

## 4. Discussion

In this study, we examined the association between anemia and measures of postpartum depression using data from 829 women in urban Malawi. We found that women with anemia are at higher risk of getting postpartum depression when measured by the “diagnostic algorithm” and, to a lesser degree, when using a cutoff score of 10 on the PHQ-9 questionnaire. These findings suggest that anemia may be more significantly associated with severe depression than with mild or moderate depression.

We also estimated a postpartum depression prevalence of 2.7% when using the “diagnostic algorithm” and an estimated prevalence of 5.1% when using the cutoff score of 10. Our estimated prevalence of anemia in the analytic sample, as determined by hemoglobin concentrations less than 110 g/L, is 37.5%, which is significantly higher than the global prevalence. Taken together, our findings indicate that both postpartum depression and anemia continue to be important health concerns in Malawi that warrant attention. Due to the potential linkages between anemia and postpartum depression, policies and programs that aim to address postpartum anemia may fulfil a dual-purpose by also contributing to the prevention of postpartum depression in women.

There were a number of limitations in our study. Due to the cross-sectional design of this study, we were not able to identify the causal direction between anemia and postpartum depression. Based on our previous literature reviews, possible causal pathways include: (1) the role of iron deficiency, whereby anemia may play a role in exacerbating emotional and psychological symptoms, such as irritability, apathy, sadness, and hypoactivity, following birth, which in turn may increase the risk of postpartum depression [23]; (2) the impact of anemia on breastfeeding and child rearing, which are known risk factors for postpartum depression [24]; (3) conversely, the role of postpartum depression in increasing the risk of anemia through postpartum nutritional deficiencies, dyspepsia, or anorexia [25]. Future research in this domain would more effectively identify the causal direction between these factors.

While we controlled for a large number of variables that might confound the association between anemia and postpartum depression, a number of other key factors were not included in our model due to data constraints. Potential missing variables include income, diet, social support, experience of intimate partner violence, HIV status, and response to the pregnancy by the father of the child, all of which have been shown to represent risk factors for postpartum depression [13]. Also, our lack of information on depression prior to birth prevents evaluation of the effects of pregnancy on the incidence of depression and on controlling for pre-birth depression status.

Moreover, our sample shows a relatively low prevalence of postpartum depression when compared to the previously identified rates in Malawi (11%) and in Africa (18%) [2,13,26]. This may be explained by differences in the definition of depression, the timing of the measurement, the location of the study area, and the relatively small sample size. Because of the small number of women in our sample with moderate and severe anemia, we had limited ability to detect potential dose-response relationships and to generalize our findings to other regions.

While we recognize the preliminary nature of the results, our study provides pioneering empirical evidence of the relationship between maternal anemia status and postpartum depression. Because postpartum depression can pose additional adverse effects, such as causing other physical and mental issues of the mother, disturbing mother-infant interaction and attachment, affecting child behavioral and cognitive development, and leading to marital distress, it is vital to screen for postpartum depression risk [27,28,29,30]. With our findings, local clinicians and midwives are able to narrow screening targets and quickly identify anemic mothers who are at higher risk of progression to depression. Luckily, most types of maternal anemia, such as the iron-deficiency anemia, folate-deficiency anemia, and vitamin B12 deficiency anemia, are often preventable and fairly easy to treat by changing or adding supplements to the mother’s diet [31,32]. According to the WHO recommendation, daily oral iron (30–60 mg) and folic acid (400 μg) supplementation throughout pregnancy can effectively reduce the risk of maternal anemia and iron deficiency [33]. Hence, the findings of this study are encouraging and suggest a feasible approach to address the issue of postpartum depression.

Our findings are not only in line with most prior studies conducted in countries such as Japan, Iran, Korea, and Saudi Arabia—suggesting a significant relationship between anemia and postpartum depression—but also shine a light on possible research forward [8,9,10,34]. It is concerning that a lack of detailed information on the specific timing of the measurements on anemia might result in inconsistent results; therefore, we suggest that future studies, if possible, should further examine the relationship across pre-, during, and post-pregnancy periods and between each trimester. Additionally, various types of measures on postpartum depression have been adopted across the existing literature, which makes it difficult to directly compare among countries; therefore, there is a need to develop a statistically valid and reliable measure for postpartum depression. Finally, as our results are not able to speak to a specific mechanism that might drive this association, future researchers should consider employing a quasi-experimental design to understand the causal relationship between anemia and postpartum depression.

## 5. Conclusions

Our findings suggest some notable variations in postpartum depression relative to the Malawian mothers’ anemia status, and strongly indicate that such variations are more significant among individuals with major depressive disorder than among those with mild or moderate depression. While several limitations emphasize the preliminary nature of these findings, the results of our study still offer new insight to health planners and practitioners into the relationship between anemia and postpartum depression and highlight the need for early screening for postpartum depression. Furthermore, we provide evidence for policymakers to promote programs that jointly address both conditions, given their potential complementarities. Finally, we recommend that future research further examines the causal relationship between anemia and postpartum depression and the validity of the measurement of major depressive disorder using the “diagnostic algorithm” in other settings.

## Figures and Tables

**Figure 1 ijerph-20-03178-f001:**
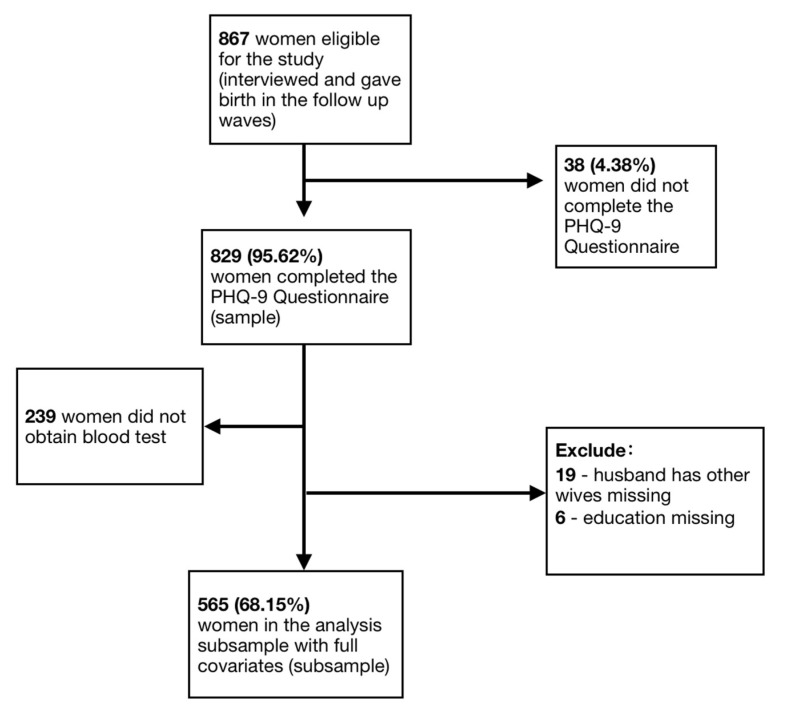
Flowchart of recruitment of final analytic sample and subsample.

**Figure 2 ijerph-20-03178-f002:**
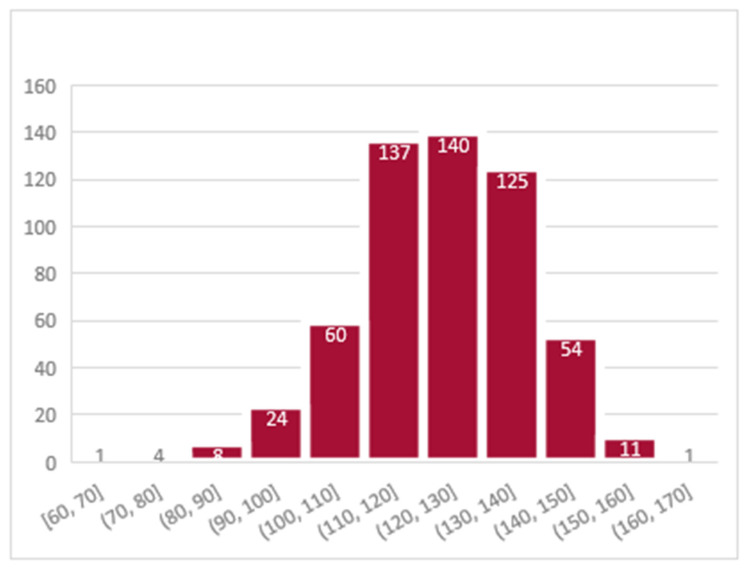
Frequency distribution of hemoglobin levels (N = 565). The histogram shows the distribution of the adjusted hemoglobin levels for the subsample of 565 women who participated in the survey of the family planning study in Malawi, completed the PHQ-9 Questionnaire, participated in blood testing, and had no missing values on the covariates.

**Figure 3 ijerph-20-03178-f003:**
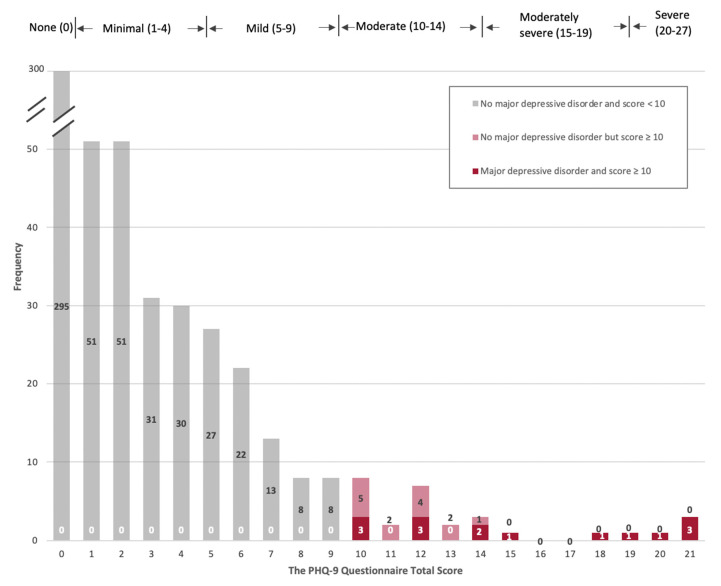
Frequency distribution of the PHQ-9 Depression Score and Major Depressive Disorder (N = 565). The histogram presents the distribution of the total scores of PHQ-9 questionnaires for a subsample of 565 women who participated in the second wave (2017) and third wave (2018) follow-up surveys of the family planning study in Malawi, completed the PHQ-9 questionnaire, and had no missing values on the covariates. Individuals who were diagnosed with major depressive disorder all scored above 10 points, but not vice versa.

**Table 1 ijerph-20-03178-t001:** Descriptive Statistics of total sample and analytic subsample.

Variables		All	Analysis Sample
		(N = 829)	(N = 565)
		n (%)	n (%)
PHQ-9 Total Score		2.10 (3.56)	2.21 (3.55)
Mean (standard deviation)			
Depression (PHQ-9 total score, categorical)			
No depression (0)		454 (54.76)	295 (52.21)
Minimal (1–4)		228 (27.50)	163 (28.85)
Mild (5–9)		110 (13.27)	78 (13.81)
Moderate (10–14)		24 (2.90)	22 (3.89)
Moderately severe (15–19)		7 (0.84)	3 (0.53)
Severe (20–27)		6 (0.72)	4 (0.71)
Major depression (shaded areas, binary)			
	Yes	23 (2.77)	15 (2.65)
	No	806 (97.23)	550 (97.35)
PHQ-9 score ≥ 10			
	Yes	37 (4.46)	29 (5.13)
	No	792 (95.54)	536 (94.87)
Age (years)			
	15–19 *	108 (13.03)	68 (12.04)
	20–24	350 (42.22)	247 (43.72)
	25–29	192 (23.16)	127 (22.48)
	30–39	179 (21.59)	123 (21.77)
Education			
	No schooling and primary *	448 (54.04)	307 (54.34)
	Secondary and higher	366 (44.15)	258 (45.66)
	Missing	15 (1.81)	0
Employment status			
	Not working *	527 (63.57)	371 (65.66)
	Working	302 (36.43)	194 (34.34)
Husband has other wives			
	Yes	47 (5.67)	31 (5.49)
	No *	751 (90.59)	534 (94.51)
	Missing	31 (3.74)	0
Multiple births			
	Yes	18 (2.17)	16 (2.83)
	No *	811 (97.83)	549 (97.17)
Total number of alive children per woman			
	Mean (standard deviation)	2.31(1.25)	2.34 (1.28)
Anemia			
	Yes	223 (26.90)	212 (37.52)
	No *	367 (44.27)	353 (62.48)
	Missing	239 (28.83)	0
Anemia severity			
	None *	367 (44.27)	353 (62.48)
	Mild	127 (15.32)	120 (21.24)
	Moderate	91 (10.98)	87 (15.40)
	Severe	5 (0.60)	5 (0.88)
	Missing	239 (28.83)	0
Hemoglobin (g/L)			
	Mean (standard deviation)	123.23 (14.84)	123.40 (14.82)

* indicates the reference group.

**Table 2 ijerph-20-03178-t002:** Chi-Square Test for Trend (One-Way ANOVA).

Variable		Summary of PHQ-9 Total Scores			
Anemia		Mean		Std. Dev.	Freq.
No Anemia		2.12		3.32	353
Mild		2.53		3.95	120
Moderate to Severe		2.12		3.89	92
Total		2.21		3.56	565
	**Sum of Squares**	**DF**	**Mean Square**	**F-Statistic**	** *p* ** **-Value**
Between groups	15.87	2	7.93	0.63	0.5345
Within groups	7110.07	562	12.65		
Total	7125.94	564	12.63		

As determined by one-way ANOVA (F_2,15.87_ = 0.63, *p* = 0.5345), there were no statistically significant differences in the PHQ-9 questionnaire mean scores among the three different groups of the independent variable, anemia (i.e., “No Anemia”, “Mild”, and “Moderate and Severe”).

**Table 3 ijerph-20-03178-t003:** Regression: Odds ratios, *p*-values, and 95% confidence intervals (CI) for correlates of major depressive disorder and moderate to severe depression (a cutoff score of 10 on the PHQ-9 questionnaire), using anemia as a binary variable.

	Major Depressive Disorder			Major Depressive Disorder			Moderate to Severe Depression		
	(Unadjusted Model)			(Model 1)			(Model 2)		
	Odds Ratio	95% CI	*p*-Value	Odds Ratio	95% CI	*p*-Value	Odds Ratio	95% CI	*p*-Value
Anemia status (g/dL)									
Yes	3.45	1.16 to 10.22	0.03 *	3.48	1.15 to 10.57	0.03 *	1.90	0.89 to 4.07	0.10
Age (years)									
20–24				0.74	0.08 to 7.04	0.79	0.85	0.22 to 3.29	0.81
25–29				0.81	0.07 to 9.36	0.87	0.58	0.11 to 2.98	0.52
30–39				0.31	0.02 to 5.73	0.43	0.54	0.08 to 3.44	0.51
Education									
Secondary and higher				2.49	0.75 to 8.31	0.14	1.20	0.52 to 2.74	0.67
Employment status									
Working				1.84	0.63 to 5.42	0.27	1.05	0.47 to 2.34	0.91
Husband has other wives									
Yes				3.45	0.65 to 18.36	0.15	2.07	0.56 to 7.64	0.28
Multiple births									
Yes				2.11	0.23 to 18.96	0.51	2.45	0.51 to 11.85	0.27
Total number of alive children per woman				1.60	0.90 to 2.81	0.11	1.34	0.88 to 2.03	0.17

* indicates the result is significant at *p* < 0.05.

## Data Availability

The data presented in this study are available on request from the corresponding author. The data are not publicly available due to the privacy of research participants.

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
