# Peer review of "Exploring the Relationship between Anemia and Postpartum Depression: Evidence from Malawi"

_ijerph, 2023, doi:10.3390/ijerph20043178_

Round 1

Reviewer 1 Report

Thank you for the opportunity to review your work.

I have some remarques and suggestions:

- Introduction:

line 40: it would be better to give a definition or explanation of 'disability adjusted life year', not every reader is familiar with this concept.

line 49-59: is there a difference between Eastern and Western countries regarding anemia and PP depression? Maybe you could add some figures of Western countries?

- Materials and methods:

It is a pity that there are no data regarding PP depression at baseline. That would be interesting to see if they were already depressed.

It is also not clear for me at which time point the data collection took place. How many weeks or months postpartum were the 3 data collections? I can image that a woman experienced other feelings at 4 weeks postpartum compared to 6 months (maybe it is also related to having a baby who cries a lot, ...).

- Results:

line 200: what could be possible causes of the lower figures for depression in your sample? Was the sample representative?

Table 3: why is the complete output presented?

Table 2 (is preceded by table 3?): is not so clear presented. It took me a while to figure out how to read this table, model 3 and 4 can be find in the supplementary information in appendix 4. 
There are a lot of references to tables, appendices, ... which makes the manscript difficult to read/follow.

- Discussion:

Do you have figures about the psyche of the pregnant women? Maybe there were already psychological problems in women who reported adepressive symptoms in postpartum, which were not related to anemia.

line 277-285: When you took the blood samples: how many weeks after the delivery? Was there still a relation between the delivery (an blood loss) and the hemoglobin values? There are several variables mentioned who could also have an influence on the hemoglobin values. So the link between PP depression and anemia is not clear for me.

Author Response

Please see the attachment, thank you!

Reviewer 2 Report

Dear Authors, 

The study deals with a very important topic therefore I was pleased to read it and thank you for the opportunity to review it. The paper is written in a correct manner, however, it is not free of errors that need to be corrected. 

-In the introduction, demonstrate the research gap for your study.

-Specify the purpose of the study or hypotheses.

-In the material and methods section, describe the inclusion criteria for the study.

-How the female-blood-questionnaire relationship was coded and identified.

-It would be prudent to separate the discussion from the conclusions and identify the strengths and limitations of the study. As the authors showed in the introduction, such studies have been conducted, so there is something to discuss.

Several interesting studies have been published in the last year, it may be worth including in the discussion.

Kwak, D.-W.; Kim, S.; Lee, S.-Y.; Kim, M.-H.; Park, H.-J.; Han, Y.-J.; Cha, D.-H.; Kim, M.-Y.; Chung, J.-H.; Park, B.; Ryu, H.-M. Maternal Anemia during the First Trimester and Its Association with Psychological Health. Nutrients 2022, 14, 3505. https://doi.org/10.3390/nu14173505

Aoki, C.; Imai, K.; Owaki, T.; Kobayashi-Nakano, T.; Ushida, T.; Iitani, Y.; N. Nakamura; Kajiyama, H.; Kotani, T. Possible effects of supplementation on postpartum depression and anemia. Medicina 2022, 58, 731. https://doi.org/10.3390/medicina58060731

-Adapt the way of citation to the requirements of MDPI - ACS Style 

Greetings!

Author Response

Please see the attachment, thank you!

Round 2

Reviewer 1 Report

Thank you for your responses. You made an important effort to improve the manuscript. I received an extensive answer regarding my comments. The way data were collected is much clearer, tabels are clearer, references were added, ...

On the other hand, I still find it difficult that there are a lot of limitations due to lack of baseline data and the wide period of collecting data. I also know that here is no solution for this problem and that further research should focus on these limitations, as you mentioned. 

Reviewer 2 Report

Dear Authors,

Thank you for making the suggested corrections, I feel satisfied. 

Greetings